bioinformatics/biochemistry/computational biology

polyproline helix, PPII, polyproline II helices, predictor, protein, secondary structure

**Author for correspondence:**
Denis C. Shields
e-mail: denis.shields@ucd.ie

# Prediction of polyproline II secondary structure propensity in proteins

Kevin T. O'Brien[1,2], Catherine Mooney[3], Cyril Lopez[1,2], Gianluca Pollastri[3,4] and Denis C. Shields[1,2]

[1]School of Medicine, [2]Conway Institute of Biomolecular and Biomedical Research, [3]School of Computer Science, and [4]Institute for Discovery, University College Dublin, Dublin, Ireland

 DCS, 0000-0003-4015-2474

*Background:* The polyproline II helix (PPIIH) is an extended protein left-handed secondary structure that usually but not necessarily involves prolines. Short PPIIHs are frequently, but not exclusively, found in disordered protein regions, where they may interact with peptide-binding domains. However, no readily usable software is available to predict this state. *Results:* We developed PPIIPRED to predict polyproline II helix secondary structure from protein sequences, using bidirectional recurrent neural networks trained on known three-dimensional structures with dihedral angle filtering. The performance of the method was evaluated in an external validation set. In addition to proline, PPIIPRED favours amino acids whose side chains extend from the backbone (Leu, Met, Lys, Arg, Glu, Gln), as well as Ala and Val. Utility for individual residue predictions is restricted by the rarity of the PPIIH feature compared to structurally common features. *Conclusion:* The software, available at http://bioware.ucd.ie/PPIIPRED, is useful in large-scale studies, such as evolutionary analyses of PPIIH, or computationally reducing large datasets of candidate binding peptides for further experimental validation.

## 1. Introduction

Polyproline II helices (PPIIHs) are an important class of secondary structure which makes up approximately 2% of the protein structure database (PDB) and are enriched in protein binding regions [1,2]. PPIIH conformations are adopted by peptides when binding to SH3, WW, EVH1, GYF, UEV and profilin domains [3,4]. They play roles in a wide variety of contexts [5]. The absence of hydrogen bonding interactions that characterize alpha helices has led to suggestions that water molecule interactions may play a role in stabilizing the helix. However,

the recent determination of a polyproline II helix structure without water molecules suggests that neighbouring amide group interactions may be sufficient to allow helix formation [6]. While proline is the key residue for PPIIH formation, other residues may impact on the stability of the helix [7] and PPIIH may be seen in sequences lacking proline [8].

While there is a database annotating experimentally defined structures of PPIIH regions [9] and several methods for assigning PPIIH regions to protein structures [10–13], there is no publicly available tool for the prediction of PPIIH based on protein sequence.

Given that many PPIIHs occur in disordered protein regions which are under-represented in structural databases, such a tool would be valuable. A neural network approach using windows of 7 and 13 residues was able to predict PPIIH secondary structure with about 75% accuracy in a dataset with redundancy elimination of sequences with more than 65% identity [14]. A support vector machine implementation using a 25% identity cut-off had an accuracy of 70% [15]. However, neither of these methods is readily available for use, and to our knowledge, no more recent methods have been developed to predict PPII helices. Accordingly, we set out to develop a predictor.

Here, we set out to develop a tool that could predict PPIIH from protein sequences, using bidirectional recurrent neural networks (BRNN). These have previously been applied successfully to predicting protein secondary structure [16–18] and other structural properties of proteins [19], and they do not require prior definition of a window size for analysis. This modelling approach has a natural representation of the peptide as a sequence, and the main advantage over other neural network architectures is the ability of distant regions of protein sequence to influence the prediction of another region. We developed a predictor, PPIIPRED, assessed its accuracy, and investigated its utility in interpreting sequence properties. PPIIPRED is available at http://bioware. ucd.ie/PPIIPRED.

# 2. Methods

The predictor was implemented using a bidirectional recurrent neural network (BRNN) architecture [20]. We trained five separate models on different partitions between the training set and validation set (which is used to assess training progression but not for tuning the model parameters). These were then ensembled to predict PPII instances in an independent test dataset. Performance was evaluated by measuring the true positive rate and false-positive rate (FPR). Receiver operating characteristic (ROC) curves were then used to plot the performance at different cut-offs ranging from 0 to 1. The count of true positives (TPs) versus the $\log 10$ of the false positives (FPs) was also visualized to evaluate performance, since the total number of FPs is much larger than TPs. See electronic supplementary material for more details.

## 2.1. Training and test datasets

Protein structures were obtained from the PDB and PISCES databases [21–23]. PISCES was used to extract redundancy reduced PDB stuctures (percentage identity $\leq 30\%$) and filtered for high-quality structures only (resolution $\leq 2.5$, $R$-value $\leq 0.25$). We used the DSSP program [24] to assign dihedral angles, and removed sequences for which DSSP does not produce an output due to missing entries or formatting errors. We defined the set of PPIIHs, applying filtering rules used in the literature [1]. We investigated both 'strict' and 'less strict' definitions. The strict criteria were
trans filtering

$$-145 < \alpha C - 70$$

dihedral filtering

$$-180 < \Psi < -160$$
$$90 < \Psi < 180$$

and

$$-105 < \Phi < -45$$

regularization filtering

$$\frac{\sum_{k=1}^{n-1} d_{k,k+1}}{n}$$

**Table 1.** Training and test dataset compositions, strict (with non-strict in parentheses).

| | sequences | PPIIH regions | PPIIH residues | non-PPIIH residues | PPIIH residues |
|---|---|---|---|---|---|
| total dataset | 9333 (10 211) | 15 112 (25 755) | 51 337 (90 249) | 2 133 861 (2 237 843) | 2.3% (3.9%) |
| training dataset | 8387 (9169) | 13 645 (23 245) | 46 382 (81 440) | 1 880 279 (2 019 181) | 2.4% (3.9%) |
| rest dataset | 945 (1040) | 1465 (2504) | 4948 (8789) | 201 969 (218 201) | 2.4% (3.9%) |

where

$$d_{k-1,k} = \sqrt{(\Psi_{i-1} - \Psi_i)^2 + (\Phi_i - \Phi_{i+1})^2}.$$

The less strict definition was identical to that for the strict definition, except that the requirement of $-105 < \Phi < -45$ was removed. Thus, dihedral angle filtering constructed a set of known PPIIH structures, using either the 'strict' or 'less strict' criteria. Each residue of every sequence in the datasets was labelled as either a PPIIH residue or a non-PPIIH residue (table 1). The number of sequences in the dataset used in training the non-strict definition is larger, we require that all sequences have at least one PPIIH region (three or more residues) for inclusion.

In addition to the amino acid sequences under investigation, the BRNN also considered alignments of related sequences to that sequence, and derived statistics of each residue of the protein sequence. Multiple sequence alignments (MSA) were extracted from the NR database (uniref 90) available in March 2014. The alignments were generated by three runs of PSI-BLAST [25] with parameter $e = 10^{-3}$ (expectation of a random hit).

IUPRED was used to calculate a 'long' disorder prediction score [26] for each residue, and espritz [27] was used to calculate the 'NMR' disorder score. We included these two disorder predictions for every residue as input. Predicted disorder may provide information not only about the protein structural state, but also about the context of the residue, since PPII helices are enriched in disordered regions [28].

Thus, the inputs to the BRNN for each protein sequence were the sequence itself, the length of the sequence, the sequence alignment, and for each residue the IUPRED (long) disorder prediction score, the espritz-NMR disorder score, and an input representing an explicit indication of the charge of the residue (1 for R or K, 0 or $-1$ for D or E). Each residue is labelled as either PPIIH or non-PPIIH. PPIIPRED predicts a score between 0 and 1 for each residue indicating the propensity for PPIIH formation. High scores indicate a higher probability of PPIIH formation.

The PPIIH dataset was split into training and test datasets, where every 10th sequence was assigned to the independent test dataset, as shown in table 1. All the tests reported in this paper were run in fivefold cross-validation, where assignment to each fold was random. The fivefold datasets were of roughly equal sizes. The training and test datasets are available in electronic supplementary material.

## 3. Algorithms

We used a BRNN to learn the mapping between inputs $\mathcal{I}$ and outputs $\mathcal{O}$ (protein sequence to a PPIIH score per residue). BRNNs have been used successfully to predict protein secondary structure [16], binding within disordered protein regions [29], bioactive peptides [30] and short linear protein binding regions [31]. They have the advantage over standard feed-forward neural networks that they can automatically find the optimal context on which to base a prediction, i.e. the number of residues that are informative to determine a property. Because of their recursive nature, BRNNs also have a relatively low number of free parameters compared to other neural networks with similar input size. See Baldi *et al.* [20] for a detailed explanation of the BRNN model, and electronic supplementary material, figure S1 which illustrates the topology.

These networks take the form

$$o_j = \mathcal{N}^{(O)}\left(i_j, h_j^{(F)}, h_j^{(B)}\right)$$

$$h_j^{(F)} = \mathcal{N}^{(F)}\left(i_j, h_{j-1}^{(F)}\right)$$

$$h_j^{(B)} = \mathcal{N}^{(B)}\left(i_j, h_{j+1}^{(B)}\right)$$

and

$$j = 1, \ldots, N,$$

where $i_j$ (respectively, $o_j$) is the input (respectively, output) of the network in position $j$, and $h_j^{(F)}$ and $h_j^{(B)}$ are forward and backward chains of hidden vectors with $h_0^{(F)} = h_{N+1}^{(B)} = 0$. We parametrize the output update, forward update and backward update functions (respectively, $\mathcal{N}^{(O)}$, $\mathcal{N}^{(F)}$ and $\mathcal{N}^{(B)}$) using three two-layered feed-forward neural networks.

## 3.1. Encoding sequence and disorder information

The input $i_j$ associated with the $j^{th}$ residue contains protein sequence information and predicted disorder information

$$i_j = (i_j^{(E)}, i_j^{(T)}),$$

where, assuming that $e$ units are devoted to sequence, and $t$ to disorder information

$$i_j^{(E)} = (i_{j,1}^{(E)}, \ldots, i_{j,e}^{(E)})$$

and

$$i_j^{(T)} = (i_{j,1}^{(T)}, i_{j,t}^{(T)}).$$

Hence $i_j$ contains a total of $e + t$ components.

We used $e = 22$: beside the 20 standard amino acids, unknown or non-standard amino acids were represented as a vector of zeroes, while the 21st input encodes the length of the sequence, and the 22nd input encodes the charge.

In a second set of tests, we used $e = 43$ where, alongside the previous 22 inputs we had a further 21 representing the frequency profile of the 20 amino acids in the MSA for the protein. The 21st input represented the frequency of gaps, which provides information about the conservation of a site, and proved helpful in preliminary tests.

In both cases, we used $t = 2$ for representing disorder information as predicted by IUPRED and espritz. Hence the total number of inputs for a given residue is $e + t = 24$ in the first representation (sequence and disorder) and $e + t = 45$ in the second representation (sequence, MSA and disorder). The output is the predicted probability of the $j$-th residue belonging to a PPIIH.

## 3.2. Training, Ensembling

Training was conducted by fivefold cross-validation, i.e. five sets of training were performed in which a different fifth of the overall set was reserved for validation purposes, i.e. to monitor the progress of the training on data not used for tuning the parameters of the models. The training set was used to learn the free parameters of the network by stochastic gradient descent. Two thousand passes through the entire training set (epochs) of training were performed for each fold, with 1920 weight updates per epoch, and the learning rate (which controls how fast the algorithm converges), starting from an initial value of 0.005, was halved whenever we did not observe a reduction of the error for more than 1000 epochs.

By the end of training, the five models of the network had errors of less 4.1% on the validation set, indicating that the networks had converged to find good local optima. We averaged the results on the five validation sets to get the overall fivefold cross-validation result. Alongside fivefold cross-validation results, we also tested the ensemble of all five models on the independent test set which we had set aside, to get an unbiased estimate of its performance. This ensemble is the final implementation of PPIIPRED.

# 4. Results

## 4.1. Neural network outperforms a proline window

Since prolines dominate many but not all PPII helices, we were interested in whether the machine learning approach was clearly outperforming a much simpler 'proline window' method that assesses the frequency of prolines in a fixed window around each residue as a direct prediction of the PPIIH state. We applied this using a sliding window method with variable window sizes. The ROC curve in figure 1$a$ shows the performance of the 'proline window' predictions. Since the rarity of PPII residues results in a typical excess of false positives over true positives at many predictor thresholds, we also investigated plots of

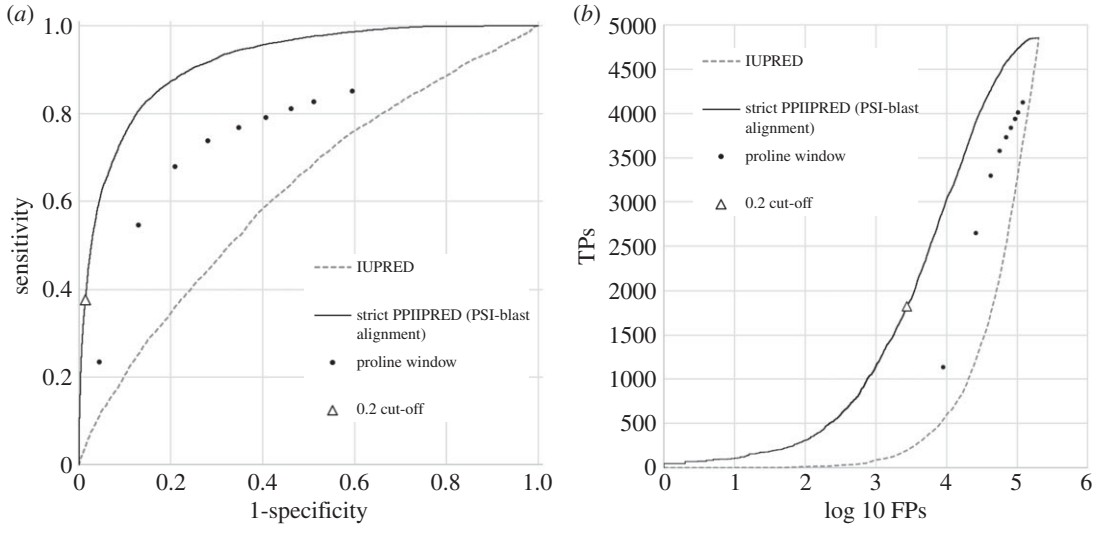

**Figure 1.** Predictive performance of PPIIPRED. Solid Line: PPIIPRED predictions. Points: predictive power of the proportion of proline alone in windows defined on a residue including between 0 and 9 residues to either side (i.e. windows of 1 to 19 in size); dashed line: prediction based on disorder prediction alone (IUPRED); black: random expectation. The point on the PPIIPRED curve corresponding to a cut-off of 0.2 is highlighted to enable cross-referencing between the images. (a) ROC curve showing predictive performance on the test dataset at various cut-offs. (b) True positives versus log10 false positives.

true positive versus log FPR (figure 1b). Although it may appear that the proline windows of size one, two or three are reasonable classifiers, the significant imbalance between TPs and true negatives (TNs) will result in a large number of false predictions when using this approach (figure 1b).

PPIIHs share some properties with disordered regions of proteins, and are often found embedded within them. Proline can interrupt alpha-helical or beta-sheet regions, and thus contribute to protein disorder. We were interested to see whether a disorder scoring method could alone provide a reasonable predictive power of the tendency to form PPIIHs. However, one standard method of disorder prediction, IUPRED, had only very weak predictive power, only modestly exceeding random predictions (figure 1).

The PPIIH predictor trained on the strict dataset and including PSI-BLAST alignments (from here on termed PPIIPRED) performed very substantially better than the disorder or proline windowing methodologies (figure 1), with an area under curve (AUC) of 0.91. At a cut-off of 0.5, it had a sensitivity of 0.86, an Mathews correlation coefficient (MCC) of 0.26 and an accuracy (Q) of 82.3. MCC and accuracy were higher at a cut-off of 0.2 (table 2). This performance compares favourably with previous methods of [14,15], although it must be pointed out that these predictors were each evaluated on different validation sets, and therefore no direct comparison is possible as their software is not made available. We noted that the strict definition of PPIIH, despite having a smaller training set of true positives, performed somewhat better than the network trained on a dataset with a less strict definition of PPIIH (table 2). We focused all further attention on the networks trained using a stricter definition of PPIIH.

## 4.2. Alignments improve predictive power

We explored the performance of the method when the alignment data is absent. While sequences alone do have reasonable predictive power, addition of alignments improves the predictions: a sensitivity of 0.23 without alignment rises to 0.38 when the alignments are included (table 2). While alignments generated on the fly by PSI-BLAST offer flexibility of analysis, in some cases better alignment accuracy can be obtained by using pre-computed alignment datasets. We wanted to check that PPIIPRED would be relatively robust to alternative means of calculating alignments. We took a set of pre-computed alignments generated for each sequence using the GOPHER approach [32]. These pre-computed alignments performed well when substituted for the PSI-BLAST alignments in the assessment of PPIIPRED, with an accuracy of 96.7 compared to 97.2 (table 2), suggesting that the approach is relatively insensitive to the particular alignment strategy adopted. However, for users studying a disordered protein region, which are often difficult to align well, it is important to check by eye that the alignment is believable, otherwise the conservation information will only mislead the

**Table 2.** Performance statistics with threshold of 0.2. See electronic supplementary material for details on the calculation of evaluation metrics. (TPs, true positives; FNs, false negatives; TNs, true negatives; FPs, false positives; Sens, sensitivity; Spec, specificity; MCC, Mathews correlation coefficient, Acc, accuracy.)

| training dataset and alignment | TPs | FNs | TNs | FPs | Sens | Spec | MCC | Acc |
|---|---|---|---|---|---|---|---|---|
| strict PPIIPRED (no alignment) | 1135 | 3720 | 1 99 367 | 2170 | 0.23 | 0.34 | 0.27 | 97.15 |
| strict PPIIPRED (PSI-Blast alignment) | 1828 | 3027 | 1 98 759 | 2778 | 0.38 | 0.40 | 0.37 | 97.19 |
| non-strict PPIIPRED (no alignment) | 2429 | 6210 | 2 04 020 | 4765 | 0.28 | 0.34 | 0.28 | 94.95 |
| non-strict PPIIPRED (PSI-Blast alignment) | 3678 | 4961 | 2 02 741 | 6044 | 0.43 | 0.38 | 0.38 | 94.94 |
| PPIIPRED (Gopher alignment) | 1418 | 2523 | 1 52 831 | 2692 | 0.36 | 0.35 | 0.34 | 96.73 |

**Table 3.** Top-scoring regions of four or more residues.

| protein | role | sequence | mean PPIIPRED score |
|---|---|---|---|
| TM175 | transmembrane protein 175 | PPPA | 0.95 |
| CCD50 | coiled-coil domain-containing protein 50 | PPPPI | 0.92 |
| RB | retinoblastoma-associated protein | PPAPPPPPPP | 0.92 |
| PALM | Paralemmin-1 | EPAP | 0.92 |
| LKAM1 | protein LKAAEAR1 | PPPA | 0.92 |
| PP12C | protein phosphatase 1 regulatory subunit 12C | PPPAE | 0.91 |
| STIP1 | stress-induced-phosphoprotein 1 | TPPPPPPPK | 0.91 |
| WASL | neural Wiskott-Aldrich syndrome protein | PPPPP | 0.91 |
| FNBP4 | formin-binding protein 4 | PPPPPPPPP | 0.91 |
| MCR | mineralocorticoid receptor | PPPPPPPP | 0.91 |

predictor, and a prediction performed without an alignment may be more accurate. Both options are available to users on the website. The user may submit multiple sequences in FASTA format within a single file, allowing a large number of predictions to be returned within one submission.

The output provides the user with numeric output of PPIIPRED scores for each residue of the user's submitted proteins. In addition, for individually submitted sequences, there is a graphic output (electronic supplementary material, figure S3) which allows the user to easily compare the findings of PPIIPRED against the backdrop of the predicted disorder in the sequence, using the IUPRED predictor.

## 4.3. High-ranking regions among human proteins

We used PPIIPRED to predict the highest scoring regions in the human proteome (cut-off = 0.5 and region length >3). As expected, the top-scoring results are dominated by proline-rich regions (table 3). However, these highest scores cannot be explained by proline composition alone, since PPPAE and PPPPP have almost identical scores. The score provided is dependent also on sequence context and evolutionary conservation, so that, for example, different scores were observed for PPPA in TM175 and LKAM1.

We were interested to explore higher confidence predicted PPII helices that were not markedly dominated by proline. While PPIIH has been proposed not to propagate beyond one sequential non-proline residue [33], we noted that one of the highest scoring proposed helices (table 3) terminates in Ala-Glu. Table 4 shows the top-scoring regions where proline is less than 40% of the motif.

Among this set of top-ranking peptides in tables 1–3, there are representations of hydrophobic (A,L,V, I,M), charged (E,K,R,D) and small polar (S,T) amino acids. By contrast, there is no representation of amino acids with bulky side-chain rings (H,F,Y,W), which may disfavour PPIIH formation. Table 5 shows top-scoring proline-free predictions. It will be of interest to experimentally examine the conformations of some of these proline-free predicted helices, particularly in the context of their larger containing proteins and protein complexes, to determine if the predictions at this edge of the comfort zone of the method have good predictive utility.

**Table 4.** Top-scoring regions with less than 40%.

| protein | role | sequence | mean PPIIPRED score |
| --- | --- | --- | --- |
| FGFR3 | fibroblast growth factor receptor 3 | MDKP | 0.8728 |
| CKAP2 | cytoskeleton-associated protein 2 | TPAV | 0.868 |
| TRML1 | trem-like transcript 1 protein | SLPA | 0.8672 |
| JIP2 | C-jun interacting protein 2 | EAPA | 0.8667 |
| PSD2 | PH and SEC7 domain-containing protein 2 | RPLL | 0.8614 |
| SMAL1 | SWI/SNF-related matrix-associated regulator of A-like protein 1 | SLPLT | 0.8605 |
| SNX14 | sorting nexin-14 | EPRS | 0.8604 |
| FBXL6 | F-box/LRR-repeat protein 6 | AAPA | 0.8546 |
| PLCB4 | 1-phosphatidylinositol 4,5-bisphosphate phosphodiesterase beta-4 | ALPS | 0.8463 |
| NIP7 | 60S ribosome subunit biogenesis protein NIP7 homologue | RPLT | 0.8456 |

**Table 5.** Top-scoring regions lacking proline.

| protein | role | sequence | mean PPIIPRED score |
| --- | --- | --- | --- |
| MDFIC | MyoD family inhibitor domain-containing protein | EALA | 0.7965 |
| RAI1 | retinoic acid-induced protein 1 | EEAA | 0.7522 |
| CTND2 | Catenin delta-2 | KKKK | 0.7486 |
| MACF1 | microtubule-actin cross-linking factor 1 | RAAS | 0.706 |
| KIF24 | kinesin-like protein KIF24 | RAAL | 0.6989 |
| NOLC1 | nucleolar and coiled-body phosphoprotein 1 | EEKL | 0.6954 |
| LIMD1 | LIM domain-containing protein 1 | LEAS | 0.68825 |
| ARHG2 | Rho guanine nucleotide exchange factor 2 | EAVA | 0.68 |
| RU17 | U1 small nuclear ribonucleoprotein 70 kDa | MEAA | 0.6691 |
| ZFHX3 | zinc finger homeobox protein 3 | RSLS | 0.6677 |

## 5. Discussion

The PPIIPRED tool offers support to those seeking to make sense of functional and evolutionary change in sequences that are likely to form polyproline II helices. This is superior to simply scanning a protein sequence by eye to identify proline-rich regions, since PPIIPRED clearly outperforms a simple proline windowing approach. It is useful to consider how reliable or interpretable the predictions may be in a typical analysis. Residues with a PPIIPRED score of greater than 0.2 account for 37% of true positives and 1.3% of false positives (figure 1b). In our test set, this translates to 1828 true positive residues and 2778 false positives. Thus, assuming that a researcher interested in predicting PPIIH within a protein was investigating a dataset similar in structural composition to the PDB test set, one false positive may be expected for every two true positives, at this cut-off, and to detect around a third of the true positives. In a practical setting, there may be a greater proportion of true positives, since many researchers interested in PPIIH are already focusing on regions of disorder, where the frequency of PPIIH is relatively high. Nevertheless, these statistics give a realistic indication of the utility of applying the predictive method to proteins in realistic conditions of interest to biologists. While this highlights the difficulties of interpreting predictions of a relatively rare structural state with modest predictive power, these predictions are of value in many biological contexts, so long as the users remain aware of the reasonable limitations of the predictions, in terms of how many false positives are typically expected for every true positive.

It is of interest to evaluate to what extent there are regions that have a high predicted PPII propensity, that also have a high alpha-helical or beta-sheet propensity. Interpretation of such findings from a machine learning predictor of PPIIH states are complex, since the dataset used in training comprises

fixed structures rather than structural ensembles, so that each residue is only found in one state. A potential consequence is that the method may to some extent use information from lack of alpha-helical or beta-sheet propensity to increase the likelihood that a residue is PPIIH. Thus, it is not clear to what extent the training algorithm of PPIIPRED may militate against residues with an alpha-helical or beta-sheet propensity, simply as a consequence of the training set provided. Careful analyses of results from structural ensembles would be required in order to tease apart these questions, and give insights into the more detailed behaviour of the PPIIPRED predictor.

Electronic supplementary material, figure S2 gives an indication of the contribution of different amino acids to human proteome predictions at various cut-offs of PPIIPRED. Clearly, the most highly predicted residues are almost all prolines, but at the cut-off of 0.2, which was previously discussed, there is a very substantial contribution of different amino acids. The preferred amino acids in these predictions match to some extent the previously known information regarding PPII propensity, with a preference among negatively charged residues for E over D in PPIIHs, previously noted in PPIIH [7]. However, the preference for methionine over leucine noted by Kentsis *et al.* [7] is not seen here, suggesting that their experimental investigation of different amino acids in the context GGxGG may not have general relevance to PPIIH formation in all contexts. In comparing other similar pairs of amino acids, a preference is also seen (electronic supplementary material, figure S2) for lysine over arginine, and for glutamine over asparagine. While glycine has a key role in PPIIH formation in the context of triple-helical collagens, it is avoided in the predicted PPIIH helices (electronic supplementary material, figure S2). Triple helical collagen structures are not well represented on the structure databases or in this training set, and are best predicted by other prediction approaches looking for the strong triplet periodicity of extended collagen regions. All PPIIHs have an exact triplet periodicity. However, the bulk of regions in this training set are short, so while it is possible that the BRNN may have used and incorporated some signal relating to short-range periodicity in refining the prediction, this is hard to assess. One potential application of PPIIPRED may be in defining candidate PPIIH regions, in which triplet periodicities of possible functional importance may be assessed, such as any amphiphilic tendencies of the helices.

Data accessibility. The 'data' is essentially the software contained in the neural networks. These are provided as a electronic supplementary material allowing researchers to download the code and run it on a Linux platform.

Authors' contributions. K.O.: lead role in the project, developing the software and implementation, data analysis, and writing the first draft of the paper. G.P.: development of the software for neural networks and input into the strategy adopted. C.M.: detailed technical support in neural network modelling and paper writing; D.C.S.: project initiation, design, interpretation and paper writing; C.L.: development of software for web analysis. All authors gave final approval for publication and agree to be held accountable for the work performed therein.

Competing interests. We declare we have no competing interest.

Funding. This work was funded through a Science Foundation Ireland (SFI) PI grant no. (08/IN.1/B1864) to D.C.S. Supported by the H2020-MSCA-RISE project IDPfun-GA no. 778247.

Acknowledgements. The authors wish to acknowledge the Irish Centre for High-End Computing (ICHEC) for the provision of computational facilities and support.

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
