## [Reviewer comments · Royal Society Open Science]

Review History

RSOS-191239.R0 (Original submission)

Review form: Reviewer 1

Is the manuscript scientifically sound in its present form?

Yes

Are the interpretations and conclusions justified by the results?

Yes

Is the language acceptable?

Yes

Do you have any ethical concerns with this paper?

No

Have you any concerns about statistical analyses in this paper?

No

Recommendation?

Accept with minor revision (please list in comments)

Comments to the Author(s)

The work described in the manuscript is timely. Left-handed PPII helices have been discovered in non-fibrillar proteins over 25 years back. They are now universally accepted as one of the major classes of protein secondary structure; PPII helices also carry distinct functional role. Still only a few attempts had been made previously to predict PPII structure. The authors have developed an algorithm aimed at predicting PPII helices and implemented it in the web server and stand alone software. The training and test datasets, as well as PPII helix definitions seem reasonable. Methodically, this is solid work, with sound approach and adequate implementation.

There are a few comments that should be addressed.

(1) Summary.

Short PPII helices are frequently found in disordered protein regions...

- PPIIs are found not only in disordered regions.

(2) Summary.

We observed an area under the curve (AUC) of 0.91 seen with an external validation set.

- statements of such level of granularity should not be present in the summary

(3) ...primary protein sequences...

- this is scientific jargon, it is not a correct term (used throughout the manuscript)

(4) Predicting PPII helices.

Only two relatively old publications are cited. Is this a full list? A more detailed discussion of such software would be appropriate; if there was no recent attempts to predict PPII, this should be specifically mentioned.

(5) It is not clear where Figure 1A is and where is 1B although one can guess. They are not marked apparently. It would also help to have a more detailed description of Tables in their headers. Some of them are rather cryptic.

(6) Presentation of results in the PPIIPred server output window is not intuitive enough. There is a colour scale bar shown there but it is not clear when a PPII helix is actually predicted - according the applied cutoff value, and what residues are included in such helix. E.g. when a sequence from the PDB entry 2BF9 is run, the results seem to show that no PPII helix was predicted, although entry 2BF9 (avian (turkey) pancreatic polypeptide) contains one of the better known examples of a long PPII helix with prolines. It would help if a more detailed description of the prediction results is added on the output page. Perhaps also a couple of test cases should be included both as part of the server and in the paper, with figure(s) showing and explaining how to interpret the output.

Review form: Reviewer 2

Is the manuscript scientifically sound in its present form?

Yes

Are the interpretations and conclusions justified by the results?

Yes

Is the language acceptable?

Yes

Do you have any ethical concerns with this paper?

No

Have you any concerns about statistical analyses in this paper?

No

Recommendation?

Accept with minor revision (please list in comments)

Comments to the Author(s)

The idea of being interested in the PPII may seem absurd or superficial. The idea of predicting PPII can therefore be considered totally irrelevant. The work carried out here shows the exact opposite, because without minimizing the difficulty of the prediction of the PPII, it demonstrates its interest.

The text is clear, effective, it lacks near nothing.

A first point is that the work of the PII (as for beta-turns and others) is also George Rose and as a reader, this ms lacks some articles that I find a little essential. Likewise, in addition to PROSS, SEGNO and other approaches exist to assign PPIIs and a little discussion would be relevant as a guide to the entire next prediction.

It would be good to put in the text the % of residues in PPII (for the reader, I had to redo the calculations).

Similarly, how can the numbers in parentheses of the non-strict all increase (the databank grows?). I probably did not understand.

Finally, a less essential question, but one that challenges. Are all PPII residues (well or poorly predicted) predicted as coils or not (by predicate approaches to secondary structures well known to the authors).

Decision letter (RSOS-191239.R0)

04-Oct-2019

Dear Dr Shields

On behalf of the Editors, I am pleased to inform you that your Manuscript RSOS-191239 entitled "Prediction of polyproline II secondary structure propensity in proteins" has been accepted for publication in Royal Society Open Science subject to minor revision in accordance with the referee suggestions. Please find the referees' comments at the end of this email.

The reviewers and handling editors have recommended publication, but also suggest some minor revisions to your manuscript. Therefore, I invite you to respond to the comments and revise your manuscript.

- Ethics statement

- Data accessibility

It is a condition of publication that all supporting data are made available either as supplementary information or preferably in a suitable permanent repository. The data accessibility section should state where the article's supporting data can be accessed. This section

should also include details, where possible of where to access other relevant research materials such as statistical tools, protocols, software etc can be accessed. If the data has been deposited in an external repository this section should list the database, accession number and link to the DOI for all data from the article that has been made publicly available. Data sets that have been deposited in an external repository and have a DOI should also be appropriately cited in the manuscript and included in the reference list.

If you wish to submit your supporting data or code to Dryad (<http://datadryad.org/>), or modify your current submission to dryad, please use the following link:
<http://datadryad.org/submit?journalID=RSOS&manu=RSOS-191239>

- **Competing interests**

- **Authors' contributions**

- **Acknowledgements**

- **Funding statement**

Because the schedule for publication is very tight, it is a condition of publication that you submit the revised version of your manuscript before 13-Oct-2019. Please note that the revision deadline will expire at 00.00am on this date. If you do not think you will be able to meet this date please let me know immediately.

To revise your manuscript, log into <https://mc.manuscriptcentral.com/rsos> and enter your Author Centre, where you will find your manuscript title listed under "Manuscripts with Decisions". Under "Actions," click on "Create a Revision." You will be unable to make your

revisions on the originally submitted version of the manuscript. Instead, revise your manuscript and upload a new version through your Author Centre.

Once again, thank you for submitting your manuscript to Royal Society Open Science and I look

forward to receiving your revision. If you have any questions at all, please do not hesitate to get in touch.

on behalf of Dr Francois Fages (Associate Editor)
 openscience@royalsociety.org

Associate Editor Comments to Author (Dr Francois Fages):

Dear authors

It is my pleasure to accept your paper with minor revisions based on the two enclosed reviews. It is nevertheless important that you address the comments made by the reviewers in your revision to take the final decision.

Best regards

Reviewer comments to Author:

Reviewer: 1

Comments to the Author(s)

The work described in the manuscript is timely. Left-handed PPII helices have been discovered in non-fibrillar proteins over 25 years back. They are now universally accepted as one of the major classes of protein secondary structure; PPII helices also carry distinct functional role. Still only a few attempts had been made previously to predict PPII structure. The authors have developed an algorithm aimed at predicting PPII helices and implemented it in the web server and stand alone software. The training and test datasets, as well as PPII helix definitions seem reasonable. Methodically, this is solid work, with sound approach and adequate implementation. There are a few comments that should be addressed.

(1) Summary.

Short PPIIHs are frequently found in disordered protein regions...

- PPIIs are found not only in disordered regions.

(2) Summary.

We observed an area under the curve (AUC) of 0.91 seen with an external validation set.

- statements of such level of granularity should not be present in the summary

(3) ...primary protein sequences...

- this is scientific jargon, it is not a correct term (used throughout the manuscript)

(4) Predicting PPII helices.

Only two relatively old publications are cited. Is this a full list? A more detailed discussion of such software would be appropriate; if there was no recent attempts to predict PPII, this should be specifically mentioned.

(5) It is not clear where Figure 1A is and where is 1B although one can guess. They are not marked apparently. It would also help to have a more detailed description of Tables in their headers. Some of them are rather cryptic.

(6) Presentation of results in the PPIIPred server output window is not intuitive enough. There is a colour scale bar shown there but it is not clear when a PPII helix is actually predicted - according the applied cutoff value, and what residues are included in such helix. E.g. when a sequence from the PDB entry 2BF9 is run, the results seem to show that no PPII helix was

predicted, although entry 2BF9 (avian (turkey) pancreatic polypeptide) contains one of the better known examples of a long PPII helix with prolines. It would help if a more detailed description of the prediction results is added on the output page. Perhaps also a couple of test cases should be included both as part of the server and in the paper, with figure(s) showing and explaining how to interpret the output.

Reviewer: 2

Comments to the Author(s)

The idea of being interested in the PPII may seem absurd or superficial. The idea of predicting PPII can therefore be considered totally irrelevant. The work carried out here shows the exact opposite, because without minimizing the difficulty of the prediction of the PPII, it demonstrates its interest.

The text is clear, effective, it lacks near nothing.

A first point is that the work of the PII (as for beta-turns and others) is also George Rose and as a reader, this ms lacks some articles that I find a little essential. Likewise, in addition to PROSS, SEGNO and other approaches exist to assign PPIIs and a little discussion would be relevant as a guide to the entire next prediction.

It would be good to put in the text the % of residues in PPII (for the reader, I had to redo the calculations).

Similarly, how can the numbers in parentheses of the non-strict all increase (the databank grows?). I probably did not understand.

Finally, a less essential question, but one that challenges. Are all PPII residues (well or poorly predicted) predicted as coils or not (by predicate approaches to secondary structures well known to the authors).

Author's Response to Decision Letter for (RSOS-191239.R0)

See Appendix A.

Decision letter (RSOS-191239.R1)

04-Dec-2019

Dear Dr Shields,

It is a pleasure to accept your manuscript entitled "Prediction of polyproline II secondary structure propensity in proteins" in its current form for publication in Royal Society Open Science. The comments of the reviewer(s) who reviewed your manuscript are included at the foot of this letter.

on behalf of Dr Francois Fages (Associate Editor)

Appendix A

UCD Conway Institute of Biomolecular and
Biomedical Research

Denis Shields PhD
Professor of Clinical Bioinformatics

Room G053, Conway Institute,
University College Dublin, Belfield, Dublin 4, Ireland

T +353 1 716 6735
denis.shields@ucd.ie
www.ucd.ie/conway

23 October 2019

To: Andrew Dunn
Royal Society of Open Science

Dear Andrew,

Thank you for the opportunity to revise our paper for Royal Society of Open Science.

Revisions for each point raised by the reviews are outlined below.

Reviewer: 1

The work described in the manuscript is timely. Left-handed PPII helices have been discovered in non-fibrillar proteins over 25 years back. They are now universally accepted as one of the major classes of protein secondary structure; PPII helices also carry distinct functional role. Still only a few attempts had been made previously to predict PPII structure. The authors have developed an algorithm aimed at predicting PPII helices and implemented it in the web server and stand alone software. The training and test datasets, as well as PPII helix definitions seem reasonable. Methodically, this is solid work, with sound approach and adequate implementation.

There are a few comments that should be addressed.

1.1

Summary.

Short PPIIHs are frequently found in disordered protein regions...

– PPIIs are found not only in disordered regions.

This has been updated to:

“Short PPIIHs are frequently, but not exclusively, found in disordered protein regions...”

1.2

Summary.

We observed an area under the curve (AUC) of 0.91 seen with an external validation set.

– statements of such level of granularity should not be present in the summary

This sentence has been replaced by:

“The performance of the method was evaluated in an external validation set.”

1.3

...primary protein sequences...

– this is scientific jargon, it is not a correct term (used throughout the manuscript)

The word “primary” has been removed where relevant and “protein” added where applicable.

1.4

Predicting PPII helices.

Only two relatively old publications are cited. Is this a full list? A more detailed discussion of such software would be appropriate; if there was no recent attempts to predict PPII, this should be specifically mentioned.

We have replaced the sentence at the end of the second introduction paragraph to read:

“However, neither of these methods is readily available for use, and to our knowledge no more recent methods have been developed to predict PPII helices. Accordingly, we set out develop a predictor.”

1.5

It is not clear where Figure 1A is and where is 1B although one can guess. They are not marked apparently. It would also help to have a more detailed description of Tables in their headers. Some of them are rather cryptic.

Figure labels have been added. Title descriptions have been added for each abbreviated columns in the tables.

1.6

Presentation of results in the PPIIPred server output window is not intuitive enough. There is a colour scale bar shown there but it is not clear when a PPII helix is actually predicted – according the applied cutoff value, and what residues are included in such helix. E.g. when a sequence from the PDB entry 2BF9 is run, the results seem to show that no PPII helix was predicted, although entry 2BF9 (avian (turkey) pancreatic polypeptide) contains one of the better known examples of a long PPII helix with prolines. It would help if a more detailed description of the prediction results is added on the output page. Perhaps also a couple of test cases should be included both as part of the server and in the paper, with figure(s) showing and explaining how to interpret the output.

A PDF document describing the output, with examples, has been added as a link on the web server. This discusses the example suggested by the reviewer as follows: “Predictions from short sequences may not be very accurate, as the training set was for sequences of 30 or more residues. For the PDB entry 2BF9, which is only 35 residues in length, the region of the short protein predicted most likely to be PPII indeed matched the known observed region in residues 2-8 (Biopolymers. 1983 22:293-304); however the scores were low (highest value 0.30) indicating weak confidence of this prediction.

The predictions are more convincing for the P53 protein, where the block of scores (maximum score 0.62) highlights the known extended region involving PPII helix formation at residues 58-91 (Proc Natl Acad Sci U S A. 2008 105: 5762–5767).

“

Reviewer: 2

Comments to the Author(s)

The idea of being interested in the PPII may seem absurd or superficial. The idea of predicting PPII can therefore be considered totally irrelevant. The work carried out here shows the exact opposite, because without minimizing the difficulty of the prediction of the PPII, it demonstrates its interest. The text is clear, effective, it lacks near nothing.

Revision 2.1:

A first point is that the work of the PII (as for beta-turns and others) is also George Rose and as a reader, this ms lacks some articles that I find a little essential. Likewise, in addition to PROSS, SEGNO and other approaches exist to assign PPIIs and a little discussion would be relevant as a guide to the entire next prediction.

We now include a reference to Rose's work at the end of the first paragraph, where the last sentence now reads as follows:

“While proline is the key residue for PPIIH formation, other residues may impact on the stability of the helix \citep{Kentsis2004} and PPIIH may be seen in sequences lacking proline \citep{Shi2002}.”

References to each of the PPIIH assignment papers have been added in paragraph 2 of the introduction:

“While there is a database annotating experimentally defined structures of PPIIH regions \citep{Chebrek2014} and several methods for assigning PPIIH regions to protein structures \citep{Srinivasan1999a,Mansiaux2011a,King1999b,Cubellis2005a}. Thus, there are good methods to identify the location of PPIIH in known structures. In contrast, there are no available tools for the prediction of PPIIH based on the protein sequence.”

Revision 2.2:

It would be good to put in the text the % of residues in PPII (for the reader, I had to redo the calculations).

This has been added to Table 1.

Revision 2.3:

Similarly, how can the numbers in parentheses of the non-strict all increase (the databank grows?). I probably did not understand.

We have clarified this point in the methods section, where we identify this aspect of the data, and the reason for it, as follows (See Methods, section 2.1):

“The number of sequences in the dataset used in training the non-strict definition is larger, we require that all sequences have at least one PPIIH region (3 or more residues) for inclusion.”

Revision 2.4:

Finally, a less essential question, but one that challenges. Are all PPII residues (well or poorly predicted) predicted as coils or not (by predicate approaches to secondary structures well known to the authors).

We agree that this is an interesting question. We have included the following as the second-last paragraph in the discussion.

“It is of interest to evaluate to what extent there are regions that have a high predicted PPII propensity, that also have a high alpha-helical or beta-sheet propensity. Interpretation of such findings from a machine learning predictor of PPIIH states are complex, since the dataset used in training comprises fixed structures rather than structural ensembles, so that each residue is only found in one state. A potential consequence is that the method may to some extent use information from lack of alpha-helical or beta-sheet propensity to increase the likelihood that a residue is PPIIH. Thus, it is not clear to what extent the training algorithm of PPIIPRED may militate against residues with an alpha-helical or beta-sheet propensity, simply as a consequence of the training set provided. Careful analyses of results from structural ensembles would be required in order to tease apart these questions, and give insights into the more detailed behaviour of the PPIIPRED predictor.”

I hope the revised version of this manuscript adequately addresses the points raised. If you have any further requests, please do not hesitate to contact me. I look forward to hearing from you,

Best Regards

Denis Shields PhD